# Upregulated Proteasome Subunits in COVID-19 Patients: A Link with Hypoxemia, Lymphopenia and Inflammation

**DOI:** 10.3390/biom12030442

**Published:** 2022-03-13

**Authors:** Enrique Alfaro, Elena Díaz-García, Sara García-Tovar, Ester Zamarrón, Alberto Mangas, Raúl Galera, Eduardo López-Collazo, Francisco García-Rio, Carolina Cubillos-Zapata

**Affiliations:** 1Respiratory Diseases Group, Respiratory Service, La Paz University Hospital, IdiPAZ, 28029 Madrid, Spain; quiquealfaro23@gmail.com (E.A.); elena.diaz.garcia.1994@gmail.com (E.D.-G.); sarugarto@gmail.com (S.G.-T.); ester.zamarron@gmail.com (E.Z.); mangasmoro@gmail.com (A.M.); raulgalera@hotmail.es (R.G.); 2Biomedical Research Networking Center on Respiratory Diseases (CIBERES), 28029 Madrid, Spain; 3Innate Immune Response Group, IdiPAZ, 28029 Madrid, Spain; elcollazo@hotmail.com; 4Faculty of Medicine, Autonomous University of Madrid, 28029 Madrid, Spain

**Keywords:** COVID-19, proteasome subunits, hypoxemia, lymphopenia, hyperinflammation

## Abstract

Severe COVID-19 disease leads to hypoxemia, inflammation and lymphopenia. Viral infection induces cellular stress and causes the activation of the innate immune response. The ubiquitin-proteasome system (UPS) is highly implicated in viral immune response regulation. The main function of the proteasome is protein degradation in its active form, which recognises and binds to ubiquitylated proteins. Some proteasome subunits have been reported to be upregulated under hypoxic and hyperinflammatory conditions. Here, we conducted a prospective cohort study of COVID-19 patients (*n* = 44) and age-and sex-matched controls (*n* = 20). In this study, we suggested that hypoxia could induce the overexpression of certain genes encoding for subunits from the α and β core of the 20S proteasome and from regulatory particles (19S and 11S) in COVID-19 patients. Furthermore, the gene expression of proteasome subunits was associated with lymphocyte count reduction and positively correlated with inflammatory molecular and clinical markers. Given the importance of the proteasome in maintaining cellular homeostasis, including the regulation of the apoptotic and pyroptotic pathways, these results provide a potential link between COVID-19 complications and proteasome gene expression.

## 1. Introduction

Coronavirus disease 2019 (COVID-19) is a new respiratory disorder caused by severe acute respiratory syndrome Coronavirus 2 (SARS-CoV-2). The clinical manifestations of COVID-19 range from asymptomatic to severe pneumonia. Growing evidence highlights the relevance of the immune response in severe COVID-19, including inflammatory cytokine storm, lymphopenia and non-T-cell response [1,2,3,4].

Lymphopenia has been frequently associated in COVID-19 patients with severe and fatal disease, indeed, it has been observed in up to 96.1% of patients with severe COVID-19 [2], and T cells are usually reduced in acute SARS-CoV-2 infection [5]. A meta-analysis found that lymphopenia is prominent in severe COVID-19, suggesting that lymphocyte counts lower than 1.5 × 10^9^/L might be useful for predicting clinical outcome [4]. Furthermore, SARS-CoV-2 infection has been proposed to cause lymphopenia, mainly through lymphocyte apoptosis and pyroptosis, and T-cell exhaustion [6,7,8].

It is well described that continuous exposure to high levels of cytokines result in a phenomenon called T-cell exhaustion. Functions of the exhausted T-cells are impaired so that inflammatory states extend without effective limitation of infection [9]. Furthermore, lymphopenia and T-cell exhaustion have been observed in recent viral pandemics such as H1N1 Influenza A [10,11], SARS-CoV [12,13] and Middle East respiratory syndrome (MERS)-CoV [14]. In fact, a higher expression of inhibitory receptors such as PD-1 and Tim-3 has been observed in PBMCs from patients with COVID-19 [3].

However, T-cell exhaustion might not be the only cause of lymphopenia in COVID-19 patients. In fact, patients with COVID-19 have elevated levels of serum lactate dehydrogenase, compared with healthy controls [15,16], which is accepted as a pyroptosis marker [17]. This type of programmed cell death, leads to an inflammatory exacerbation state in which cytoplasmic contents and pro-inflammatory cytokines, including IL-1β, are released [18]. In this way, pyroptosis links the two signature events in SARS-CoV-2 infection, i.e., lymphocyte depletion and hyperinflammation. In addition, recent reports show that RNA viruses promote the activation of the NLRP3 inflammasome, which leads to pyroptotic cell death as a consequence of gasdermin D (GSDMD) cleavage by caspase-1 [19].

Incidence of lymphopenia is related to sepsis pathophysiology [20,21] and is involved in clinical complications after heart transplant [22]. In septic patients, ARDS, cytokine storm and lymphopenia are concomitant, which is similar to the process occurring in severe COVID-19 patients [23,24]. Interestingly, in a recent report, Xue and colleagues identified a set of genes that were upregulated in septic patients with hypoxaemic phenotype. Among those, the authors found immune-related genes as NLRP3 and STAT3, which have been previously described in studies of sepsis and hypoxia pathogenesis. Strikingly, Xue et al. also reported the upregulation of some proteasome subunits, for which the relevance in hypoxemia and hyperinflammation remains to be elucidated [25]. Indeed, 20S proteasome has been described not only at the basis of several diseases but also has been highlighted as an early prognostic biomarker for sepsis development mainly related to lymphocyte apoptosis [26].

The proteasome is made up of two subcomplexes: a catalytic core particle (also known as the 20S proteasome) and one or two terminal regulatory particles. The 20S proteasome is composed of four heptameric rings, which consist of seven structurally related α and β subunits, displaying an α1-7, β1-7 setup [27,28]. The 19S regulatory particle recognize ubiquitylated proteins, deubiquitylate them for recycling of ubiquitin, and then unfolds and translocate them into the interior of the catalytic particle for degradation [29]. Ubiquitylation is, along with phosphorylation, one of the main regulatory modifications proteins can undergo [30]. In this context, the importance of the ubiquitin-proteasome system (UPS) in the regulation and cell signaling of immune cells has been described [31]. In response to viral infection, the UPS plays a key role in regulating the activity and stability of various key proteins as MAVS, TRAF3 or IKKα/β [32,33,34], which are involved in the activation of interferon response pathway and NF-κB signaling [35]. Additionally, UPS is implicated in the regulation of inflammation, with ubiquitylation acting as an important post-translational modification for NLRP3 regulation [36]. Furthermore, in most cells, oxidative stress, hypoxia or inflammatory cytokines are stimuli that lead to an elevated production of immunoproteasomes, which increase the production of peptides for the presentation on MHC class I molecules [37]. In addition to antigen processing, immunoproteasomes influence cytokine production and T cell differentiation, survival and function [38]. In fact, several viruses target the UPS to facilitate their entry by endocytosis [39,40]. Other viruses mimic ubiquitin ligases in order to shut down cellular defense proteins [41]. Additionally, the importance of UPS for efficient infection of Coronavirus has been reported [42]. In this line, MERS-CoV, SARS-CoV-1 and SARS-CoV-2 contain their own deubiquitynase-like enzyme, Nsp3-PLpro, which can target and dysregulate type-I interferon signaling [43,44,45].

Therefore, evidence suggests that the proteasome could play a key role during SARS-CoV-2 infection and disease complications. Here, we studied the expression of certain subunits of the proteasome core and regulatory particles in peripheral blood mononuclear cells (PBMCs) and their association with lymphopenia, hypoxemia and inflammation in COVID-19 patients.

## 2. Materials and Methods

### 2.1. Study Participants

A total of 20 healthy control (HC) subjects and 44 patients hospitalized for COVID-19 were consecutively recruited from La Paz University Hospital, Madrid, Spain. Inclusion criteria for the COVID-19 group were a positive result on a RT-PCR assay of an upper respiratory tract specimen for SARS-CoV-2; associated abnormalities or infiltrates on chest X-ray or chest CT; active fever or documented fever within 48 h or antipyretic use; and hypoxemia, defined as a room air oxygen saturation of less than 92% or requirement of supplemental oxygen. Exclusion criteria included aged younger than 18 years, onset of COVID-19 symptoms more than 14 days before hospital admission, concomitant systemic bacterial of fungal infection, history of immunodeficiency or neutropenia (absolute neutrophil count <1500/mm^3^), active neoplasia, history of severe pulmonary disease requiring home oxygen therapy or mechanical ventilation, history of current systemic autoimmune or auto-inflammatory disease, or previous therapy with long-term oral corticosteroids, anti-IL-1, anti-IL-6 or anti-TNF.

The COVID-19 pneumonia patients were treated according to institutional recommendations without drug restrictions. Blood samples for study measurements and hospital-based blood testing were obtained at the admission day and 7 days after. The ratio of arterial oxygen partial pressure (PaO_2_) to fractional inspired oxygen (FiO_2_) was also determined at day 1. Exploratory endpoints were mortality at day 60, admission to the ICU for intubation and mechanical ventilation and duration of hospitalization.

The study was approved by the local Ethics Committee (PI-4087) and informed consent was obtained from all the participants.

### 2.2. Peripheral Blood Mononuclear Cell Isolation

Peripheral blood mononuclear cells (PBMCs) from COVID-19 patients or HC were isolated using Ficoll-Paque Plus (Amersham Bioscience, Uppsala, Sweden) gradient by centrifugation (1500 rpm for 20 min at 24 °C). PBMCs were removed from the interphase and washed two times in PBS.

### 2.3. Hypoxia and PX-478 In Vitro Models

PBMCs were isolated from blood samples of healthy volunteers. After isolation, 2 × 10^6^ PBMCs/well were cultured in M6 plates in RPMI medium supplemented with 100 U/mL Penicillin and 100 μg/mL Streptomycin. Cells were cultured for 16 h in standard conditions (37 °C, 21% O_2_, 5% CO_2_) or using a specific hypoxic chamber (Oxycycler C42 from Biospherix, Parish, NY, USA) which can maintain 9% O_2_. For the PX478 inhibition assay, cells were treated with 30 μM of PX478 (an HIF1α inhibitor; MedKoo Biosciences Inc., Morrisville, NC, USA) [46,47] for 16 h.

### 2.4. Lipopolysaccharide (LPS) Stimulation Model

PBMCs were isolated from blood samples of healthy volunteers and COIVD-19 patients. After isolation, 2 × 10^6^ PBMCs/well were cultured in M6 plates in RPMI medium supplemented with 100 U/mL Penicillin and 100 μg/mL Streptomycin. Cells were cultured for 16 h in standard conditions (37 °C, 21% O_2_, 5% CO_2_) or with or without LPS (10 ng/mL) for 16 h.

### 2.5. mRNA Isolation and Quantification by Quantitative PCR

Then, mRNA was isolated from PBMCs using the High Pure RNA Isolation Kit (Roche Diagnostics, Basel, Switzerland), following the manufacturer’s protocol. After quantification, complementary DNA (cDNA) was obtained by reverse transcription of 1 ug RNA using the High-Capacity cDNA Reverse Transcription kit (Applied Biosystems, (Waltham, Massachusetts, USA). Then, human PSMA4,5, PSMB1-10, PSMD11,14, PSME1-3, HIF-1α, NF-κB, NLRP3, ASC, CAS-1, GSDMD, STAT3, c-FOS, c-JUN and 18S expression levels were measured by real-time quantitative polymerase chain reaction (qPCR) using the CFX96 Touch Real-Time PCR Detection System (Bio-Rad Laboratories, (Hercules, CA, USA) and QuantiMix Easy SYG kit from Biotools (Madrid, Spain) with specific primers, which are listed in Appendix A, and then synthesized, desalted, and purified by Eurofins Scientific SE (Luxembourg). The cDNA copy number of each gene of interest was determined using a six-point standard curve and results were normalized to the expression of the house keeping gene 18S.

### 2.6. Flow Cytometry

Peripheral blood mononuclear cells (PBMCs) were treated following a standard protocol using the Transcription Factor Buffer Set (BD Biosciences, Franklin Lakes, NJ, USA) and stained with antibodies against CD14 (14A-100T, Inmunostep S.L., Salamanca, Spain) and NLRP3 (030-712, MiltenyiBiotec, Madrid, Spain). Appropriate isotype controls were used for each experiment. After staining for 30 min at 4 °C in the dark, cells were acquired using a BD FACSCalibur flow cytometry from BD Biosciences (Franklin Lakes, NJ, USA), and the collected data were analyzed using FlowJo vX.0.7 software (FlowJo, LLC, Franklin Lakes, NJ, USA). The gating strategy is depicted in Appendix A.

### 2.7. Statistical Analysis

Data are presented as mean and standard error of the mean. The means were compared using an unpaired t-test with Welch’s correction. For more than two groups, differences in mean were analysed by ANOVA test and multiple comparisons were assessed by Tukey’s test. Correlations were calculated with Spearman’s correlation test. Differences were considered significant at *p* < 0.05. The analyses were conducted using Prism 8.0 software (Graph Pad, Hercules, California, USA).

## 3. Results

### 3.1. Characteristics of the Study Subjects

COVID-19 pneumonia patients and healthy control subjects were homogeneous in sex (50% vs. 68% males, *p* = 0.243), age (52 ± 9 vs. 57 ± 13 years, *p* = 0.125) and body mass index (27.3 ± 5.6 vs. 29.7 ± 6.8 kg/m^2^, *p* = 0.405). Detailed clinical characteristics of patients with COVID-19 pneumonia are shown in Table 1. Initially, hypoxemia was treated with oxygen therapy, which was administered via the nasal canula, Venturi mask or high-flow nasal oxygen, or other non-invasive ventilation according to its severity and patient tolerance. In two patients (4.5%), respiratory failure progressed, requiring intubation and mechanical ventilation, and two patients (4.5%) died in the 60-day follow-up period.

### 3.2. COVID-19 Patients Overexpress Certain Proteasome Subunits

We examined the mRNA expression of some proteasome subunits in PBMCs from COVID-19 patients and healthy subjects (HC). Regarding the expression of 20S subunits in the α- and β-ring the mRNA expression of PSMA4, PSMA5, PSMB2, PSMB9, PSMB10 significantly increased in COVID-19 patients (COV) when compared to healthy control (HC). Moreover, PSMB1, PSMB4, PSMB8 were not overexpressed in COVID-19 patients PBMCs (Figure 1A–C). Interestingly, PSMB5, PSMB6, and PSMB7 also did not reveal a significant difference between COVID-19 patients and HC (Appendix A), suggesting an overexpression of the immunoproteasome rather than constitutive proteasome. In addition, we assessed some subunits from the 19S and 11S regulatory particles and found that PSME1 and PSMD14 were significantly elevated in COVID-19 patients in comparison to HC, while the other subunits assessed did not reach significance (Figure 1D and Appendix A).

### 3.3. Proteasome Subunits Gene Expresion Are Related to Hypoxia

Severe COVID-19 patients might develop hypoxemia, which often requires oxygen therapy. In line with this, we determined the Hypoxia-inducible factor 1-alpha (HIF-1α) mRNA expression in PBMCs from COVID-19 patients to corroborate the hypoxemia condition in these patients (Appendix A). Strikingly, proteasome-subunit mRNA expression was strongly associated with HIF-1α mRNA expression in patients (Figure 2A–D and Appendix A). These results suggest an association between proteasome mRNA expression and hypoxemia induced by SARS-CoV-2 infection.

In order to verify the contribution of HIF1-α to the upregulation of proteasome subunits in COVID-19 patients, we exposed PBMCs from healthy volunteers to hypoxic conditions in combination with PX-478 (S-2-amino-3-[4V-N, N-bis (2-chloroethyl) amino]-phenyl propionic acid N-oxide dihydrochloride), which suppresses constitutive and hypoxia-induced levels of HIF1α (Appendix A). The mRNA levels from all of the proteasome subunits studied (unless PSMB5, PSMB6) increased under hypoxic condition, and this increment was impaired when the cells were treated with PX-478 (Figure 3A–D and Appendix A).

### 3.4. Some Proteasome Subunits Are Related to Poor Clinical Outcome

Severe COVID-19 patients are characterized by lymphopenia development during the first days of infection. Therefore, we explored the association between proteasome subunits expression and lymphocyte count in patients. Patients with high PSMA4 and PSMA5 mRNA expression suggested more acute lymphopenia, more severe hypoxemia and higher concentrations of C-reactive protein and ferritin in plasma (Table 2).

### 3.5. Some Proteasome Subunits Are Associated with Inflammatory Markers

Given the acute inflammation and lymphopenia in COVID-19 patients, we tested whether the gene expression of proteasome subunit might be involved in the inflammatory pathways leading to cell death. First, we observed that PBMCs from patients tended to overexpress nuclear factor kappa B (NF-κB) and NLRP3 when compared to HC although not reaching statistical significance (Appendix A). Interestingly, all the studied proteasome subunits were positively correlated with NF-κB. Besides, the majority of proteasome subunits studied (unless PSMB9 and PSME1) positively correlated with NLRP3 mRNA expression in COVID-19 patients (Figure 4A and Appendix A).

In this line, we evaluated NLRP3 protein expression in CD14^+^ monocytes (gated strategy in Appendix A). Our data suggested that patients characterized by elevated expression of the proteasome subunits PSMA5; PSMB1,2,4; PSMB5, PBMB8, had higher levels of NLRP3 protein expression than patients characterized by low proteasome subunits mRNA expression (Figure 4B and Appendix A). Furthermore, NLRP3 inflammasome components were upregulated in COVID-19 patients compared to HC (Appendix A). In fact, we observed a direct correlation between the mRNA expression of the majority of proteasome subunits studied and the mRNA expression of members of the NLRP3 inflammasome complex such as apoptotic speck-like protein containing a caspase recruitment domain (ASC) and caspase-1 (CAS-1) (Figure 4A and Appendix A). Finally, the mRNA expression of the proteasome subunits studied (unless PSMB8 and PSME2) were also positively correlated with gasdermin D (GSDMD) (Figure 4A), the main effector molecule for the lytic and highly inflammatory form of programmed cell death known as pyroptosis.

### 3.6. Proteasome Subunitswere Related with STAT3 Cascade

Furthermore, STAT3 was identified to be upregulated in the context of intensive inflammation and hypoxemia, and its potential role in COVID-19 pathogenesis has been recently reviewed [48,49]. Here, we reported the mRNA upregulation of STAT3 together with its well-known downstream effectors c-FOS and c-JUN in COVID-19 patients compared with healthy controls (Appendix A). A positive correlation was observed between all the proteasome subunits studied and STAT3. In addition, some proteasome subunits correlated with c-FOS and c-JUN (Figure 5 and Appendix A).

### 3.7. Proteasome Subunits Upregulation May Be a Cause and Not a Consequence of Inflammation

Finally, in order to clarify whether inflammation is a cause or consequence of proteasome upregulation in COVID-19, patients we treated HC and COVID-19 patients PBMCs with LPS (10 ng/mL) during 16 h. Our in vitro data suggested that inflammation may not have an impact on proteasome subunit expression (Appendix A). Therefore, considering the proteasome subunits overexpression data in COVID-19 patients, we speculate a possible role of proteasome subunits mRNA expression in the inflammatory condition of these patients.

## 4. Discussion

Severe COVID-19 patients are well characterized by hypoxemia, a reduced number of T cells and inflammatory storm during the first days after SARS-CoV-2 infection [2,50]. Additionally, there is a wide range of evidence supporting the idea that proteasome activity play a role in the inflammatory response [51]. Indeed, ubiquitylation maintains a balance between the activation and inhibition of the immune system response due to infection [52]. In our study, we assessed the expression of 16 genes from the catalytic core and regulatory particles of the 20S proteasome. Our data showed an increased expression in seven of the subunits of proteasome (PSMA4,5; PSMB2,9,10; PSMD14 and PSME1) in PBMCs from COVID-19 patients, suggesting a potential increase in proteasome activity, since other studies have already related an overexpression of proteasome subunits with a higher proteasome activity [53,54]. Besides, many studies have attempted to elucidate the mechanisms underlying the regulation of proteasome subunit expression, leading to the identification of multiple transcription factors which in turn are regulated by several interrelated pathways [55,56,57,58,59]. This suggests that the regulation of proteasome subunit expression is intricate and complex, and supports the fact that only seven of the proteasome subunits studied are upregulated in COVID-19 patients, while the others are not. Consequently, further studies are required in order to elucidate the mechanism underlying proteasome subunits expression in a physiological and COVID-19 pathological context.

Both PSMA4 and PSMA5 are part of the α-rings of the catalytic core proteasome [60], which comprise the outer rings of the 20S proteasome [61]. We observed PSMA4 and PSMA5 overexpression in COVID-19 patients. In agreement with these findings, PSMA5 mRNA expression was elevated in patients with sepsis [25]. Furthermore, PSMB1-10 are part of the β-rings of the catalytic core proteasome (20S) [28], which comprise the inner rings of the 20S proteasome. The β-rings form a proteolytic chamber where only the subunits encoded by PSMB5, PSMB6 and PSMB7 genes demonstrate hydrolytic activity [62,63]. Furthermore, β-type proteasome subunits PSMB8, PSMB9 and PSMB10, which are homologous to PSMB5, PSMB6 and PSMB7, respectively, form the immunoproteasome, a type of proteasome with a specialized role in processing intracellular antigens for presentation to the immune system [64]. Interestingly, we only observed an increase in PSMB9 and PSMB10, suggesting mediation of the immunoproteasome in COVID-19 pathology. Curiously, mutations in genes encoding for immunoproteasome subunits have been related to the development of proteasome-associated auto-inflammatory syndromes in mouse models [65] and humans [66]. In addition, PSMD11 and PSMD14 are part of the lid of the 19S regulatory particle [28]. Our data showed an increase in PSMD14 expression. As is known, PSMD14 encodes the metalloisopeptidase Rpn11, which plays an essential role in the de-ubiquitylation of captured substrates to facilitate their degradation, which constitutes the main biochemical activity of 19S lid [67]. Besides, PSMD14 is implicated in viral replication and pathogenesis in hepatitis B [68]. Moreover, PSME1-3 are subunits of the 11S (also called PA28 or REG) regulatory particles. PSME1 and PSME2 assemble to form a complex (also called proteasome activator PA28alpha/beta) that binds the ends of the 20S proteasome and dramatically accelerates the generation of a subset of MHC class I-presented antigenic peptides. Our data revealed the upregulation of PSME1 in COVID-19 patients. Interestingly, the role of 11S proteasome during viral infection has already been reported [69,70].

Our data suggested proteasome subunits overexpression to be related to the hypoxic condition in COVID-19 patients. In agreement, some proteasome subunits have been previously reported to increase under hypoxaemic conditions [25,71,72,73]. Thus, we hypothesised that hypoxia might play a role in the upregulation of proteasome subunits in COVID-19 patients, according to the patient and in vitro model data. Other authors have highlighted the role of 20S proteasome in septic patients suggesting its involvement in immunological activity [74]. Moreover, in a recent study, 20S proteasome serum content was related to requirement of oxygen therapy in COVID-19 patients [75]. Along with hypoxia development, COVID-19 severity is tightly bounded to hyperinflammatory state.

Our data revealed a relation of the proteasome subunits with inflammation markers. Similarly, ubiquitin-proteasome axis has been described to be dually essential for inflammation via the NF-κB pathway and NLRP3 inflammasome [76]. On the one hand, all proteasome subunits studied correlated with NF-κB and an LPS in vitro model showed no effect on proteasome subunits expression, suggesting that proteasome upregulation might be implicated in COVID-19 inflammatory condition development. On the other hand, we identified a direct relation between the majority of proteasome subunits studied and NLRP3 inflammasome components (NLPR3, ASC, CAS-1). The ubiquitin-proteasome system (UPS) is involved in the regulation of NLRP3 and ASC stability by selective ubiquitilation [36,77]. Rodrigues and colleagues demonstrated NLRP3 inflammasome is activated in response to SARS-CoV-2 infection [78]. Furthermore, NLRP3 inflammasome activation drives cells into pyroptosis, a process leading to cell-death by mean of gasdermin D [79], which may be the prominent cause of lymphopenia, and high levels of inflammation. Overall, we hypothesized that proteasome subunits upregulation is related to inflammation and pyroptosis, highlighting a potential connection between UPS regulatory activities and COVID-19 clinical features. Upregulation of proteasome has also been described in critically ill septic patients and trauma patients, both sharing extensive inflammatory conditions [74].

Concomitantly to pyroptosis, lymphopenia may also be related with hyperinflammation, which is able to induce lymphocyte exhaustion and apoptosis. Although apoptotic cell death was not assessed in this study, we related proteasome subunits with STAT3. Recent reports show that the IL-6/STAT3 pathway can drive the upregulation of the transcription of TGF-β [80], which is related to T-cell exhaustion and mediates apoptosis of virus-specific CD8^+^ T cells [81], although it has been typically described as an anti-apoptotic pathway [82]. Furthermore, activated STAT3 can induce P53 transcription, enhancing COVID-19 patients’ lymphopenia [83]. Interestingly, the IL-6/STAT3 axis could contribute to an exacerbated immune response and COVID-19 complications such as thrombosis and lung fibrosis [49]. Proteasome and its subunits also play a direct role in apoptosis. Indeed, the overexpression of proteasome subunits has been identified to protect different lung and colon cancer cells from apoptosis [54,84,85,86].

In this line, we suggested that hypoxia might upregulate proteasome subunits mRNA expression, which could be implicated in lymphocyte cell-death. Many studies discuss that proteasome inhibitors are capable to induce apoptosis; indeed, they are frequently used in the treatment of cancer [87,88]. Furthermore, an inhibition of proteasome has been described to induce apoptosis in human monocyte-derived dendritic cells, suggesting a protective role of proteasome in the immune system [89]. This protective role is supported by a proteasome-dependent degradation of proteins such as NLRP3 and P53. However, other studies have demonstrated a requirement of proteasomes during apoptosis in non-cycling and differentiated cells [90]. In addition, evidence has suggested that viruses and their infected hosts exploit the proteasomal degradation machinery to promote efficient viral replication and restrict invading viruses in the infected host [42,91]. In regard, proteasome inhibitors have been demonstrated to be efficient in limiting viral life cycle, including in SARS-CoV [92,93]. Conversely other authors report that the proteasome inhibitor can also enhance viral disease [94]. In the context of COVID-19, Longhitano et al. hypothesized the therapeutic value of proteasome inhibitors [95]. The issue of whether proteasome subunits overexpression exerts a protective or pathogenic role remains to be elucidated.

The study has several limitations which we acknowledge. Firstly, the study is limited to mRNA expression quantification although other studies have related mRNA expression of proteasome subunits with a higher proteasome activity [53,54]. Secondly, the procedures with infected cells were restricted due to local biologically secure conditions. Thirdly, we have not considered the possible impact on proteasome subunits expression of the differences in PBMCs subpopulation described in COVID-19 patients. Therefore, the results obtained allow us to hypothesize involvement between the pathogenic pathways and limit the mechanistic and causative description of the processes involved.

In this study, we explored the proteasome subunits in COVID-19 patients to determine the potential association between hypoxemia, lymphopenia and hyperinflammation. Collectively, these findings suggested a link between proteasome subunits expression and the clinical manifestations in COVID-19 patients. In this line, a recent study by Wendt et al. also suggests a relation between proteasome expression and the hyperactivation of the immune system and tissue damage in COVID-19 patients [75]. Future studies should focus on the proteasome activity in SARS-CoV-2 infection severity, which might corroborate our findings on the relationship between proteasome activity and lymphopenia, hypoxemia and inflammation. (Figure 6). Additional work is needed to corroborate the causative relations of the upregulation of proteasome subunits and COVID-19 severity.

## Figures and Tables

**Figure 1 biomolecules-12-00442-f001:**
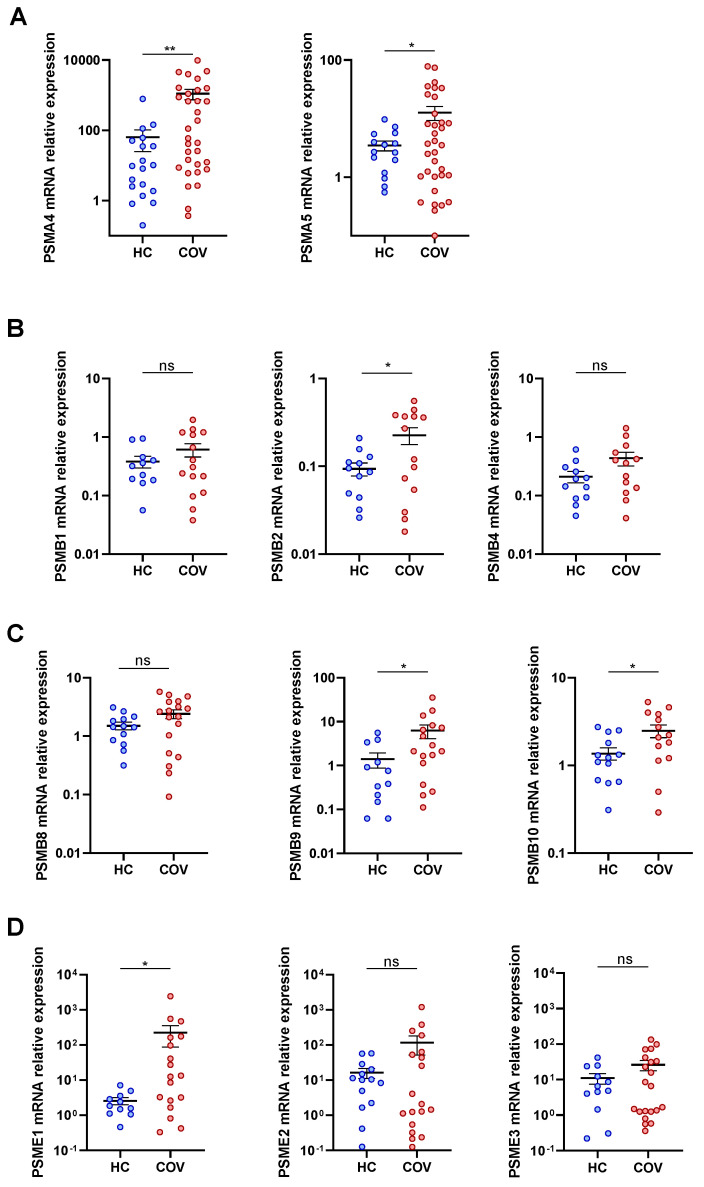
Upregulation of proteasome subunits in COVID-19 patients. mRNA expression in PBMCs from healthy controls (HC) compared to COVID-19 patients (COV) of (**A**) 20S proteasome alpha ring subunits: PSMA4 (HC *n* = 20; COV *n* = 32), PSMA5 (HC *n* = 15; COV *n* = 34); (**B**) 20S proteasome beta ring non-catalytic subunits: PSMB1 (HC *n* = 11; COV *n* = 15), PSMB2 (HC *n* = 12; COV *n* = 14), PSMB4 (HC *n* = 12; COV *n* = 13); (**C**) 20S immunoproteasome beta ring catalytic subunits: PSMB8 (HC *n* = 13; COV *n* = 18), PSMB9 (HC *n* = 12; COV *n* = 17), PSMB10 (HC *n* = 13; COV *n* = 14); and (**D**) 11S proteasome subunits: PSME1 (HC *n* = 11; COV *n* = 18), PSME2 (HC *n* = 14; COV *n* = 19), PSME3 (HC *n* = 12; COV *n* = 21). Mean differences were analysed using unpaired Student’s *t*-test analysis with Welch’s correction. Error bars: standard error of the mean. *: *p* < 0.05, **: *p* < 0.01.

**Figure 2 biomolecules-12-00442-f002:**
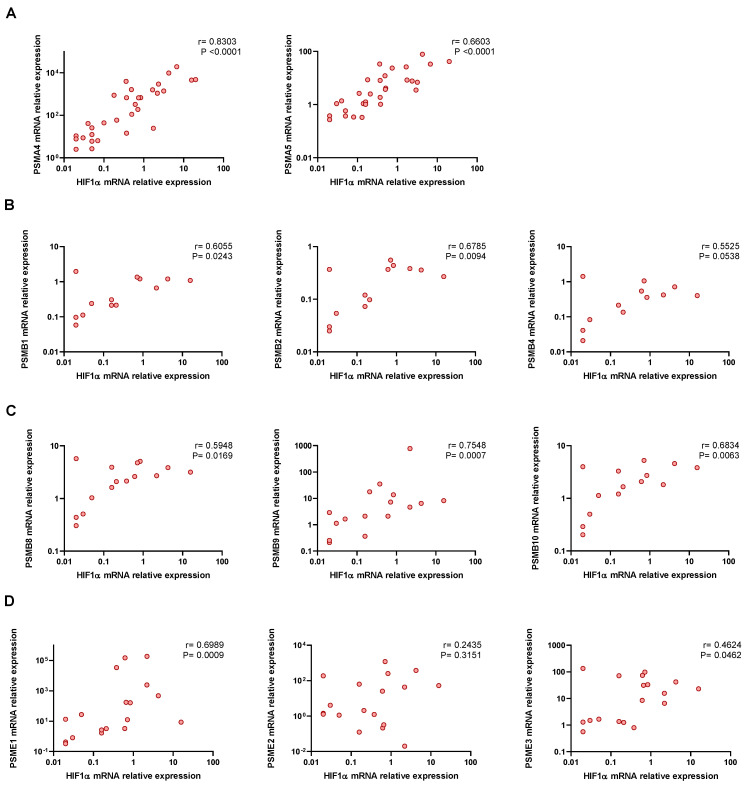
Hypoxia-inducible factor 1-alpha correlation with proteasome subunits. Correlation of hypoxia-inducible factor 1-alpha (HIF-1α) mRNA expression and mRNA expression of (**A**) 20S proteasome alpha ring subunits: PSMA4 (*n* = 31), PSMA5 (*n* = 32); (**B**) 20S proteasome beta ring non-catalytic subunits: PSMB1 (*n* = 14), PSMB2 (*n* = 14), PSMB4 (*n* = 13); (**C**) 20S immunoproteasome beta ring catalytic subunits: PSMB8 (*n* = 16), PSMB9 (*n* = 17), PSMB10 (*n* = 15); and (**D**) 11S proteasome subunits: PSME1 (*n* = 19), PSME2 (*n* = 19), PSME3 (*n* = 19). Correlations were analysed using Spearman’s analysis. *p*-value and Spearman’s correlation coefficient (ρ) are shown.

**Figure 3 biomolecules-12-00442-f003:**
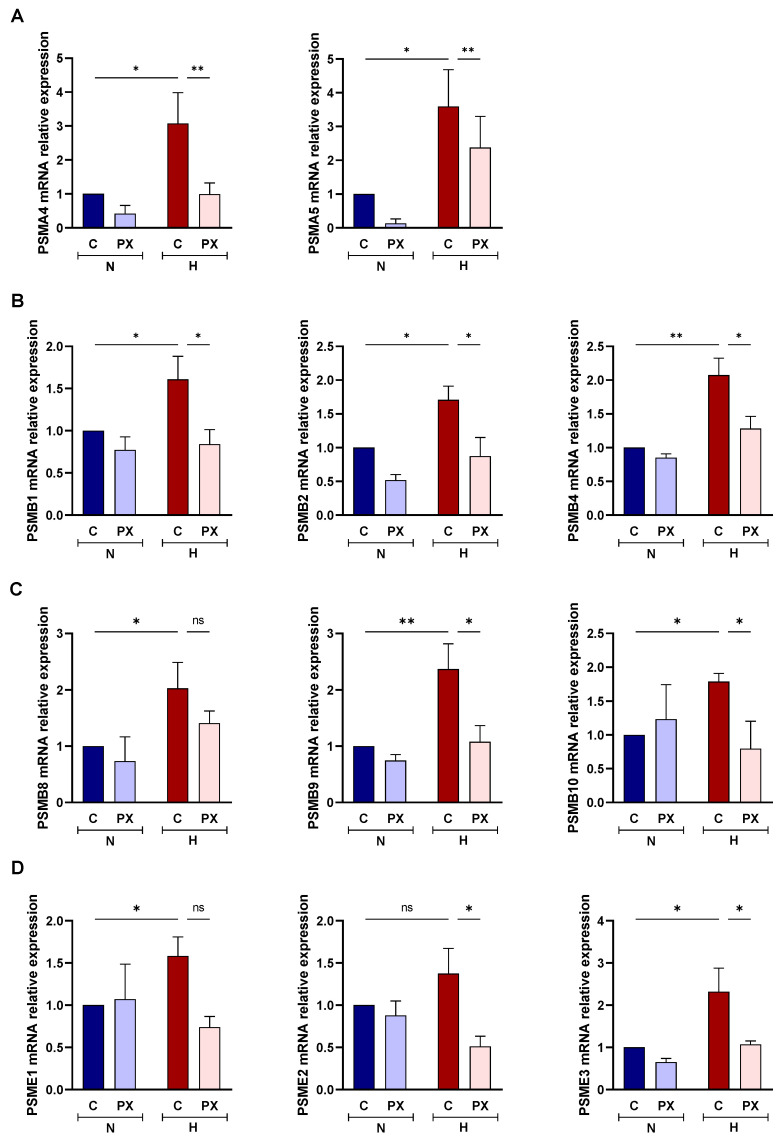
In vitro model of the effect of hypoxia on proteasome subunits expression. mRNA expression of proteasome subunits from healthy volunteers PBMCs (*n* = 5) cultured in normoxia (N) or hypoxia (H) and treated with PX478 (PX) or untreated (C). (**A**) 20S proteasome alpha ring subunits: PSMA4, PSMA5; (**B**) 20S proteasome beta ring non-catalytic subunits: PSMB1, PSMB2, PSMB4; (**C**) 20S immunoproteasome beta ring catalytic subunits: PSMB8, PSMB9, PSMB10; and (**D**) 11S proteasome subunits: PSME1, PSME2, PSME3. Mean differences were analysed using mixed-effects analysis with Bonferroni’s multiple comparison test. Error bars: standard error of the mean. *: *p* < 0.05, **: *p* < 0.01.

**Figure 4 biomolecules-12-00442-f004:**
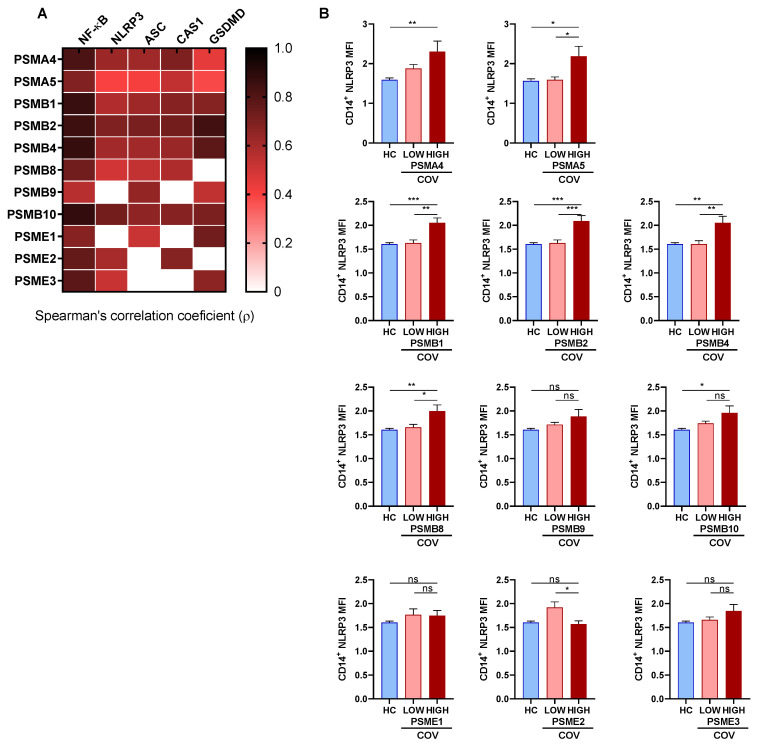
Proteasome subunits relationship with inflammation components. (**A**) Heatmap represents Spearman’s correlation coefficient (ρ) between proteasome subunits mRNA expression and genes related to inflammation mRNA expression. White gaps represent non-significant correlations, while the rest of correlations are significant (*p* < 0.05). (**B**) NLRP3 expression in monocytes analysed by flow cytometry in monocytes from healthy controls (HC), and COVID-19 patients (COV) presenting LOW or HIGH expression of proteasome subunits. Mean differences were analysed using one-way ANOVA analysis with Tukey’s multiple comparisons test. Error bars: standard error of the mean. *: *p* < 0.05, **: *p* < 0.01, ***: *p* < 0.001.

**Figure 5 biomolecules-12-00442-f005:**
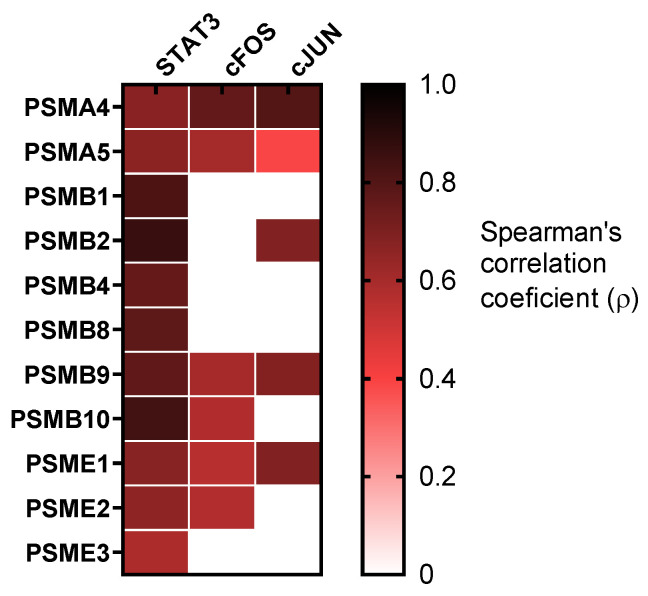
Proteasome subunits relationship with STAT3 pathway genes. Graph represents Spearman’s correlation coefficient (ρ) between proteasome subunits mRNA expression and STAT3, cFOS and cJUN. White gaps represent non-significant correlations, while the rest of correlations are significant (*p* < 0.05).

**Figure 6 biomolecules-12-00442-f006:**
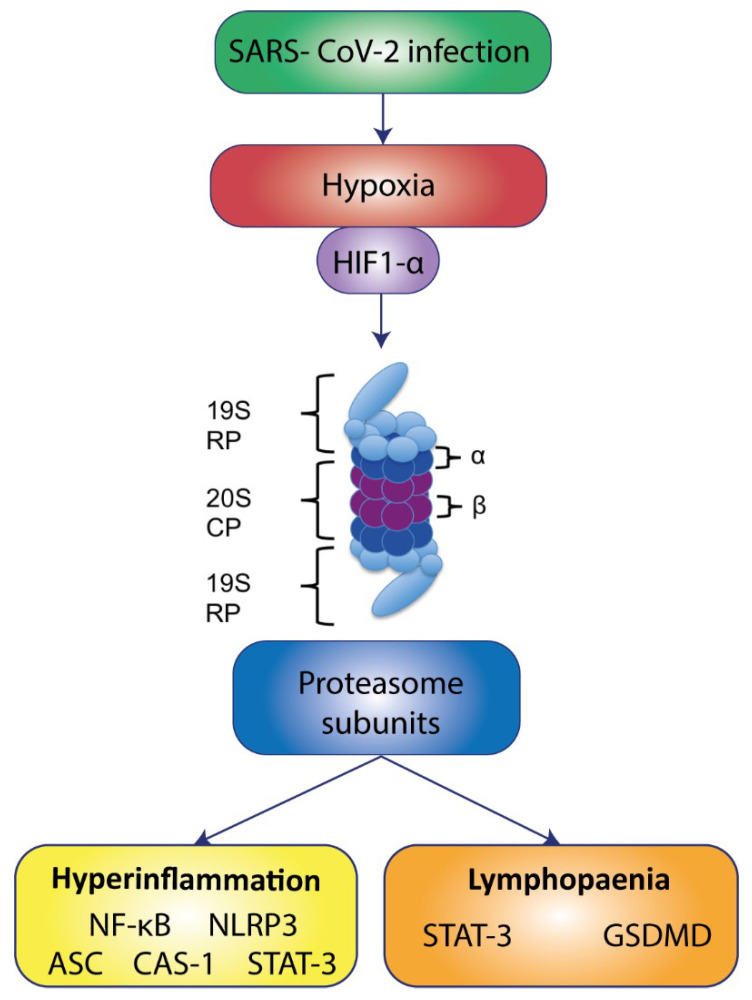
Proteasome might play an important role in COVID-19 pathogenesis. SARS-CoV-2 infection induces hypoxia which, we suggested, leads to overexpression of proteasome subunits in PBMCs from patients with COVID-19, potentially leading to an overactivation of the ubiquitin-proteasome system. The pleiotropic roles of proteasome in immune biology include regulation of master proteins such as, NF-κB, NLRP3, GSDMD and STAT3, which have been related to COVID-19 complications, mainly lymphopenia and hyperinflammation.

**Table 1 biomolecules-12-00442-t001:** General characteristics of the COVID-19.

Age, year		57 ± 13
Sex, Male/Female		30/14
Days since onset of symptoms		7.7 ± 4.0
Symptoms at admission		
	Cough	28 (64)
	Active fever	32 (73)
	Dyspnea	31 (70)
	Myalgia	15 (34)
	Sputum production	8 (18)
	Chest tightness	3 (7)
	Headache	8 (18)
	Fatigue	12 (27)
	Anorexia	5 (11)
	Nausea	3 (7)
	Diarrhea	7 (16)
	Chest pain	5 (11)
	Anosmia	6 (14)
Comorbidities		
	Hypertension	12 (27)
	Coronary artery disease	3 (7)
	Diabetes mellitus	13 (30)
	Obesity	10 (23)
	Chronic lung disease	4 (9)
	Chronic kidney disease	1 (2)
	Hypothyroidism	2 (3)
Smoking history		
	Current, *n* (%)	31 (70)
	Former, (%)	3 (7)
	Never, *n* (%)	10 (23)
Pneumonia severity scores		
	CURB-65	0.84 ± 0.88
	Fine risk class	2.4 ± 1.3
Laboratory findings		
	PaO_2_, mmHg	65.0 ± 25.9
	PaO_2_/FiO_2_ ratio	282.5 ± 113.5
	PaCO_2_, mmHg	36.8 ± 14.7
	White cell count, 10^3^ cells/µL	6.74 ± 4.41
	Neutrophils, 10^3^ cells/µL	4.90 ± 3.10
	Lymphocytes, 10^3^ cells/µL	1.31 ± 2.04
	Eosinophils, 10^3^ cells/µL	0.02 ± 0.03
	Platelets, 10^3^ cells/µL	227.7 ± 101.8
	Hemoglobin, g/dL	13.9 ± 4.7
	C-reactive protein, mg/L	68.8 ± 57.7
	Aspartate aminotransferase, U/L	48.8 ± 38.5
	Alanine aminotransferase, IU/L	43.7 ± 29.6
	Υ-Glutamyltransferase, IU/L	84.8 ± 81.0
	Bilirubin, µmol/L	0.51 ± 0.28
	Albumin, g/L	4.34 ± 1.82
	Ferritin, ng/mL	839.1 ± 899.4
	Lactate dehydrogenase, U/L	304.2 ± 163.6
	D-dimer, ng/mL	697.4 ± 677.4
	Fibrinogen, mg/dL	672.1 ± 270.9
Evolution results		
	Duration of hospital admission, days	9.9 ± 5.6
	ICU admission, *n* (%)	2 (4.5)
	*Exitus*, *n* (%)	2 (4.5)

**Table 2 biomolecules-12-00442-t002:** Clinical parameters of COVID-19 patients grouped by proteasome subunits expression.

Gene	Lymphocytes (×10^3^ Cells/mm^3^)			PaO_2_/FiO_2_ Ratio			C-Reactive Protein (mg/L)			Ferritin (mg/mL)		
	Low	High	*p*	Low	High	*p*	Low	High	*p*	Low	High	*p*
PSMA4	1.018 ± 0.293	0.740 ± 0.168	0.0084	320.0 ± 92.90	232.9 ± 105.0	0.028	46.97 ± 32.63	90.42 ± 67.20	0.044	486.1 ± 532.2	1361 ± 1158	0.024
PSMA5	1.035 ± 0.476	0.734 ± 0.164	0.0498	302.2 ± 75.82	229.5 ± 88.98	0.0429	50.36 ± 34.32	94.5 ± 66.77	0.0470	473.8 ± 475.6	927.6 ± 471	0.0339
PSMB1	0.983 ± 0.306	0.655 ± 0.199	0.056	288.3 ± 100.8	247.5 ± 82.38	0.438	41.93 ± 38.4	47.1 ± 29.8	0.791	707.7 ± 684.4	876.5 ± 696.4	0.659
PSMB2	0.983 ± 0.306	0.626 ± 0.207	0.048	288.3 ± 100.8	245.7 ± 88.81	0.441	41.93 ± 38.4	47.20 ± 32.2	0.796	707.7 ± 684.4	796.3 ± 711.2	0.824
PSMB4	1.030 ± 0.380	0.784 ± 0.360	0.301	295.7 ± 106.5	247.7 ± 90.97	0.407	48.25 ± 35.95	37.51 ± 30.14	0.576	772.3 ± 621.6	775.3 ± 734.7	0.994
PSMB8	0.957 ± 0.287	0.919 ± 0.528	0.164	284.3 ± 92.6	262.6 ± 95.12	0.663	42.57 ± 35.08	47.81 ± 29.88	0.763	812.0 ± 683.0	747.5 ± 672.7	0.857
PSMB9	0.873 ± 0.317	0.976 ± 0.471	0.623	302.0 ± 85.14	244.4 ± 88.27	0.209	52.13 ± 35.46	38.77 ± 26.52	0.423	804.9 ± 690.5	707.1 ± 639.2	0.777
PSMB10	0.798 ± 0.272	0.963 ± 0.456	0.441	296.2 ± 91.72	241.6 ± 85.97	0.283	58.45 ± 34.26	34.71 ± 29.01	0.202	936.0 ± 654.0	705.3 ± 708.5	0.541
PSME1	0.920 ± 0.311	0.885 ± 0.489	0.867	285.6 ± 89.74	249.2 ± 86.46	0.409	56.75 ± 37.74	42.46 ± 33.01	0.422	614.1 ± 618.9	795.3 ± 687.2	0.576
PSME2	0.831 ± 0.381	0.974 ± 0.423	0.491	289.4 ± 93.14	245.9 ± 81.42	0.325	58.65 ± 37.17	40.77 ± 32.65	0.312	641.8 ± 621.7	770.8 ± 691.2	0.691
PSME3	0.939 ± 0.370	0.866 ± 0.443	0.728	294.6 ± 91.56	241.2 ± 79.86	0.223	52.24 ± 32.69	46.47 ± 38.63	0.743	779.6 ± 626.1	648.2 ± 687.5	0.686

Clinical parameters (lymphocyte count, PAFI ratio, C-reactive protein concentration and ferritin concentration) comparison in COVID-19 patients grouped by high or low expression of proteasome subunits (PSMA4/5, PSMB1/2/4/8/9/10 and PSME1-3). Data are mean ± standard deviation. *p*-values are shown. Differences were considered significant *p* < 0.05.

## Data Availability

The data that support the findings of this study are available from the corresponding authors upon reasonable request. Individual level participant data will not be made available to others due to privacy concerns.

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
