# Peer review of "Upregulated Proteasome Subunits in COVID-19 Patients: A Link with Hypoxemia, Lymphopenia and Inflammation"

_biomolecules, 2022, doi:10.3390/biom12030442_

Round 1

Reviewer 1 Report

This version of the manuscript is substantially improved. I don't think the authors adequately wrestle with why some subunits of proteasomes are increased while others are not (it would be a good addition to the discussion), but the authors have improved the completeness of the study to adequate levels.

Author Response

Point-by-point Reviewer 1.

This version of the manuscript is substantially improved.

We highly appreciate the reviewer’s comment.

I don't think the authors adequately wrestle with why some subunits of proteasomes are increased while others are not (it would be a good addition to the discussion), but the authors have improved the completeness of the study to adequate levels.

Thanks for the comment, following the reviewer’s suggestion we have added a new paragraph regarding why some subunits of proteasomes are increased while others are not in the discussion section. Please find below the new sentences: 

“Besides, many studies have attempted to elucidate the mechanisms underlying the regulation proteasome subunits expression, leading to the identification of multiple transcription factors which in turn are regulated by several interrelated pathways [1-5]. This suggests that proteasome subunits expression regulation is intricate and complex, and supports the fact that only 7 of the proteasome subunits studied are upregulated in COVID-19 patients, while the others are not. Consequently, further studies are needed in order to elucidate the mechanism underlying proteasome subunits expression in a physiological and COVID-19 pathological context”.

References:

  1. Motosugi, R.; Murata, S. Dynamic Regulation of Proteasome Expression. Front Mol Biosci 2019, 6, 30, doi:10.3389/fmolb.2019.00030.
  2. Xu, H.; Fu, J.; Ha, S.W.; Ju, D.; Zheng, J.; Li, L.; Xie, Y. The CCAAT box-binding transcription factor NF-Y regulates basal expression of human proteasome genes. Biochim Biophys Acta 2012, 1823, 818-825, doi:10.1016/j.bbamcr.2012.01.002.
  3. Vilchez, D.; Boyer, L.; Morantte, I.; Lutz, M.; Merkwirth, C.; Joyce, D.; Spencer, B.; Page, L.; Masliah, E.; Berggren, W.T.; et al. Increased proteasome activity in human embryonic stem cells is regulated by PSMD11. Nature 2012, 489, 304-308, doi:10.1038/nature11468.
  4. Vangala, J.R.; Dudem, S.; Jain, N.; Kalivendi, S.V. Regulation of PSMB5 protein and beta subunits of mammalian proteasome by constitutively activated signal transducer and activator of transcription 3 (STAT3): potential role in bortezomib-mediated anticancer therapy. J Biol Chem 2014, 289, 12612-12622, doi:10.1074/jbc.M113.542829.
  5. Radhakrishnan, S.K.; Lee, C.S.; Young, P.; Beskow, A.; Chan, J.Y.; Deshaies, R.J. Transcription factor Nrf1 mediates the proteasome recovery pathway after proteasome inhibition in mammalian cells. Mol Cell 2010, 38, 17-28, doi:10.1016/j.molcel.2010.02.029.

Reviewer 2 Report

ACCEPT as it is. revisions were performed and reality has to be published. 

Author Response

We highly appreciate the reviewer's comment.